# Growth-coupled overproduction is feasible for almost all metabolites in five major production organisms

Axel von Kamp[1] & Steffen Klamt[1]

Computational modelling of metabolic networks has become an established procedure in the metabolic engineering of production strains. One key principle that is frequently used to guide the rational design of microbial cell factories is the stoichiometric coupling of growth and product synthesis, which makes production of the desired compound obligatory for growth. Here we show that the coupling of growth and production is feasible under appropriate conditions for almost all metabolites in genome-scale metabolic models of five major production organisms. These organisms comprise eukaryotes and prokaryotes as well as heterotrophic and photoautotrophic organisms, which shows that growth coupling as a strain design principle has a wide applicability. The feasibility of coupling is proven by calculating appropriate reaction knockouts, which enforce the coupling behaviour. The study presented here is the most comprehensive computational investigation of growth-coupled production so far and its results are of fundamental importance for rational metabolic engineering.

[1] ARB Group, Max Planck Institute for Dynamics of Complex Technical Systems, Sandtorstrasse 1, Magdeburg 39106, Germany. Correspondence and requests for materials should be addressed to S.K. (email: klamt@mpi-magdeburg.mpg.de).

The shift from a petrochemical to a bio-based and sustainable production of chemicals and fuels remains as a major global challenge of humanity in the twenty-first century. Diverse commercial compounds are currently produced in fermentation processes including commodity chemicals, polymers, biofuels, pharmaceuticals, nutritional supplements and so on.[1–3] To further optimize existing and to develop new fermentation processes, metabolic engineering emerged as an enabling technology. It combines experimental and theoretical approaches to engineer cell factories with maximal performance.[1,3,4] Computational modelling has become an important method for metabolic engineering, not only to gain deep insights into properties and production capabilities of metabolic networks[5] but also to identify rational metabolic intervention strategies for the design and optimization of microbial production organisms.[6]

One key design principle that has become particularly relevant for metabolic engineering and computational strain design over the past decade is to couple cellular growth with the production of a desired metabolite. The central goal is to make the desired metabolite a mandatory by-product of growth and its production thus an integral part of the organism's metabolic function (Fig. 1). In this way, growth of the organism becomes a driving force of production. Without coupling, the functionality that is needed for enhanced production may easily be lost from a production strain that adapts to a higher growth rate as this functionality usually poses a burden on the organism.[7] Furthermore, when a growth-coupled strain has been designed, it is possible to improve its production capabilities through adaptive laboratory evolution by selecting for maximum growth.[8–12]

OptKnock[13] was the first optimization method proposed for computing reaction deletion strategies to couple the production of a metabolite to cellular growth. This method can be seen as the origin of a variety of developed strain design methods for growth-coupled product syntheses.[6,14–16] In all these methods, growth-coupled product synthesis demands that mutant strains are forced to produce the desired metabolite to be able to grow with maximal growth rate or to be able to grow at all (with any rate). Using growth coupling as the design principle, a variety of mutant strains has successfully been constructed. Examples for *E. coli* are strains for the production of lactate[8], ethanol from a mixture of glucose and xylose[17] as well as glycerol[10], isobutanol[18], 1,4-butanediol[19], malonlyl-CoA[20], fatty acids[21] and itaconic acid[22]. In *Saccharomyces cerevisiae*, mutant strains have been designed for the production of 2,3-butanediol[23] and succinate[12].

However, growth coupling is not *per se* possible for every metabolite. Feasibility of growth-coupled product synthesis has recently been investigated from a theoretical point of view[24]. The authors first distinguish between weak and strong coupling and then derive criteria for the feasibility of (weakly or strongly) growth-coupled production in a given metabolic network. Weak coupling means that a sufficiently high product yield is achieved if the cell grows with maximal or close-to-maximal biomass yield (similar to OptKnock-related methods mentioned above). In contrast, strong coupling demands more, it additionally requires that production must also occur even without growth (Fig. 1). In other words, strong coupling means that substrate uptake already enforces the production of the desired metabolite. The derived criteria for feasibility of weak and strong coupling are based on elementary (flux) vectors, a generalization of elementary (flux) modes to the inhomogeneous case[25]. As a concrete example, a small-scale model of the central metabolism of *E. coli* (89 metabolites and 107 reactions) was taken to examine whether the production of each metabolite can be coupled to growth[24]. This test could be achieved rather easily because the model was small enough to quickly compute all its elementary

vectors[24]. Briefly, the necessity to calculate all elementary vectors for evaluating the criteria derives from the fact that not only an elementary vector needs to be found that supports the desired growth-coupled production but that it is also necessary to prove that there is a way to disable all other elementary vectors whose potential operation would break growth-coupled production. The surprising result was that growth-coupled product synthesis is possible for all metabolites under aerobic conditions and most metabolites in the anaerobic case[24]. This raises the question whether the same result can also be obtained in a full genome-scale model and whether growth-coupled production is possible for metabolites from other parts of the metabolism as well. An additional question is to what degree the growth-coupled synthesis of metabolites is feasible also in other relevant production organisms.

The aim of this work is therefore to investigate the feasibility of growth-coupled product synthesis in genome-scale metabolic models of five representative production organisms. The chosen species have been used as cell factories in numerous biotechnological applications, covering prokaryotes and eukaryotes as well as heterotrophic and photoautotrophic organisms. These organisms and their associated established genome-scale metabolic models are *E. coli* (iJO1366; ref. 26), *S. cerevisae* (iMM904; ref. 27), the Gram-positive bacterium *Corynebacterium glutamicum* (iJM658; ref. 28), the filamentous fungus *Aspergillus niger*[29] and the cyanobacterium *Synechocystis* sp. PCC 6803 (ref. 30).

As the computation of elementary vectors has the same complexity as the computation of elementary modes[25], this will, despite recent algorithmic advances[31,32], typically be impractical in genome-scale metabolic networks with many inputs and outputs. Therefore, the criteria for growth-coupled product synthesis[24] cannot be directly applied to the models above. One possibility to circumvent the need for calculating all elementary vectors is to search directly for a single combination of knockouts that disables product yields below a given threshold while ensuring that production and growth yields above their respective thresholds remain feasible[24]. Such an intervention strategy can be computed as a constrained minimal cut set (cMCS); if (and only if) a cMCS with these properties exists, then strong coupling is possible. A cMCS comprises a set of reactions that need to be knocked out to enforce coupling. A procedure for the direct calculation of cMCS has been described earlier[33,34] and will be applied in a modified form here. The main difference in the application of this procedure in the present study is that it is sufficient to find any cMCS to prove that growth coupling is possible. For proving coupling, the size of the cut set does not matter, whereas it was a central aim in the original application to enumerate smallest cMCS[33,34]. In contrast to other works, in all calculations we will focus on strong coupling (Fig. 1), as it demands coupling under all conditions even if the cell does not behave growth optimal.

Using our developed algorithmic pipeline, we demonstrate that suitable intervention strategies for growth-coupled overproduction exist for almost all metabolites in all five organisms investigated. These results are of fundamental importance as they show that growth-coupled product synthesis is indeed a widely applicable design principle for rational metabolic engineering.

## Results

**Computational framework for testing feasibility of coupling.** The organisms and associated models for which the feasibility of strong coupling was examined are listed in Table 1. Two of the organisms, *A. niger* and *C. glutamicum*, have a very limited

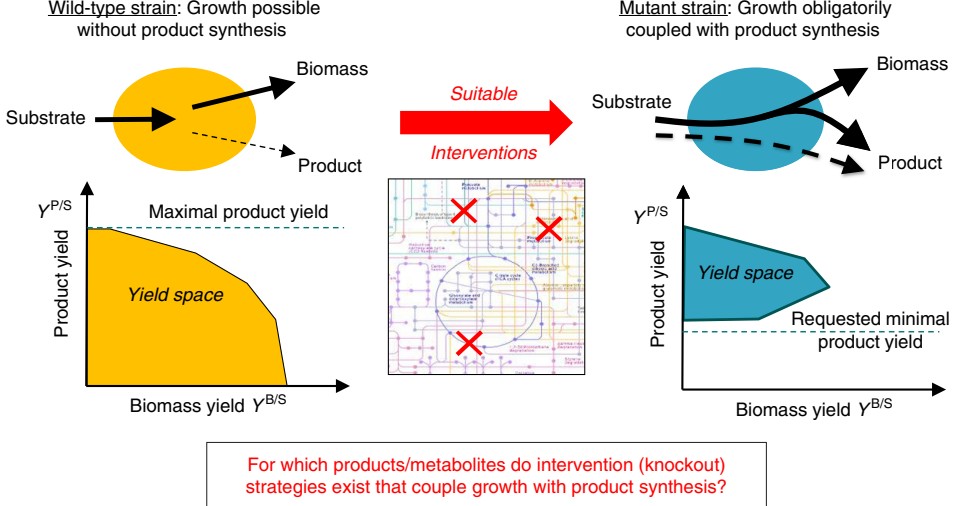

**Figure 1 | Design of production strains with growth-coupled product synthesis.** The yield space shows the projection of all feasible steady-state flux distributions in the network with respect to their biomass yield and product yield. Herein we demand strong coupling, meaning that in the mutant strain to be constructed, growth without product synthesis is not possible anymore (production without growth is allowed). All (remaining) flux distributions in the mutant strain reach a product yield above a demanded threshold.

capability for anaerobic growth, while *E. coli* and *S. cerevisiae* can grow aerobically as well as anaerobically. The growth of these four heterotrophic organisms was simulated on glucose minimal medium. The fifth organism, *Synechocystis* sp. PCC 6803, is photoautotrophic and was simulated with light as limited 'substrate' together with an unlimited uptake of $CO_2$. Specific details on model configurations used in the calculations can be found in the Methods section. Briefly, all models were provided with an unlimited supply of inorganic compounds (via the respective exchange reactions) necessary for growth while uptake of the substrate glucose (photons in case of *Synechocystis* sp. PCC 6803) was limited to known maximal values. Outflows of typical organic (for example, fermentation) products for the given organism were left open and thus have to be accounted for when the cut sets were calculated. Importantly, to ensure that the calculated knockout strategies (cut sets) have a high degree of biological relevance, reactions were set to be irrepressible and can thus not be knocked out if they do not correspond to enzyme-catalysed biochemical reactions (for example, spontaneous reactions or pseudo reactions representing transport processes) or if no associated gene is known (in models where gene-reaction mappings were available). Overall, the percentage of irrepressible reactions in the respective models is significant and reaches up to 34.5% (in *E. coli*) of the operative reactions (Table 1).

In every model it was then tested for each organic metabolite producible from the substrate whether a suitable knockout strategy (cMCS) exists such that growth and production of the metabolite can be strongly coupled. For each candidate metabolite an exchange reaction for this compound was temporarily added to the model and calculations were done for three different levels of demanded minimal product yield, specifically 10, 30 and 50% of the maximum yield for the respective metabolite (see Methods section). The 10% level was chosen to check if strong coupling is in principle possible, whereas the 50% level provides information about whether the metabolite can be produced with a high yield under coupling. The intermediate level 30% was included to see in more detail how the feasibility of strong coupling changes with increasing minimum product yield. To test whether growth-coupled production is possible, it was attempted to calculate a cMCS for each metabolite and given minimum yield.

The calculation of such a cMCS is a computationally hard problem. Based on earlier developments, we therefore built an algorithmic pipeline to determine effectively such cMCS by solving dedicated Linear Programming (LP) and Mixed Integer Linear Programming (MILP) problems (see Methods section). For a given metabolite and yield level, the algorithm seeks to either find a cMCS proving coupling or to disprove feasibility of coupling. If the MILP problem cannot be solved nor its infeasibility be determined within the given time limit (see Methods section), then it is not possible to decide if strong coupling is feasible. This happened only in relatively few of the considered cases (see below).

**Feasibility of coupling.** Figure 2 shows the results of the computations for *E. coli* and *S. cerevisiae*; these two models were simulated under aerobic and anaerobic conditions (detailed results of all calculations can be found in the Supplementary Data 1). As a major finding for both organisms, it can be seen that for over 96% of all metabolites producible from the substrate glucose strong coupling is feasible under aerobic conditions for all three levels of coupling. For the 10% yield level, suitable interventions for growth coupling were found for even more than 99% of the substrate-producible metabolites. These results were unexpected, as they demonstrate an almost unrestricted feasibility of growth-coupled strain design for producing any of the native metabolites in *E. coli* and yeast, even when taking the large number of irrepressible reactions into account (Table 1). Figure 2 shows that the fraction of metabolites that can be coupled drops (slightly) with increasing minimum yield, which can be expected because with increasing yield it becomes more difficult to ensure with knockouts that sufficient flux is forced through the reactions that participate in production. However, only for a very minor percentage of the metabolites (<4% in both organisms), feasibility of coupling cannot be determined any more when demanding a minimum yield of 50%, instead of 10% of the maximal yield. For all cases, where a cMCS inducing strong coupling under aerobic conditions was not found in reasonable time, a final proof of infeasibility of coupling could not be given by the solver within the set time limit (see Methods section). Hence, the percentages shown for aerobic growth in Fig. 2 should

**Table 1 | Details of the models used in this study.**

| Organism (model name) | Internal metabolites | Reactions (operative) | Repressible reactions (operative) | Irrepressible reactions (operative) | Substrate uptake limit (mmol gDW$^{-1}$h$^{-1}$) | ATP maintenance demand (mmol gDW$^{-1}$h$^{-1}$) | Reference |
|---|---|---|---|---|---|---|---|
| *E. coli* (iJO1366) | 1,805 | 2,582 (1,761) | 1,414 (1,154) | 1,168 (607) | 15 (glucose) | 3.15 | 26 |
| *S. cerevisiae* (iMM904) | 1,228 | 1,577 (1,217) | 1,020 (822) | 557 (395) | 10 (glucose) | 1 | 27 |
| *C. glutamicum* (iJM658) | 984 | 1,065 (647) | 788 (510) | 277 (137) | 5.4 (glucose) | 3.2 | 28 |
| *A. niger* | 1,037 | 1,280 (1,094) | 961 (877) | 319 (217) | 2 (glucose) | 1.9 | 29 |
| *Synechocystis* sp. PCC 6803 | 518 | 594 (594) | 584 (584) | 10 (10) | 100 (photons) | — | 30 |

ATP, adenosine triphosphate.

Operative reactions are those that can carry a flux under the considered constraints and growth conditions (minimal medium). Irrepressible reactions (for example, exchange reactions, transporters, technical or pseudo reactions) are not allowed to be knocked out. The substrate for all organisms is glucose, except for *Synechocystis* sp. PCC 6803, which absorbs photons. ATP maintenance is a technical reaction that represents the ATP requirements for non-growth-associated maintenance processes. Where such a reaction was provided by the respective models, the lower limit of its flux is set to the value shown in the table.

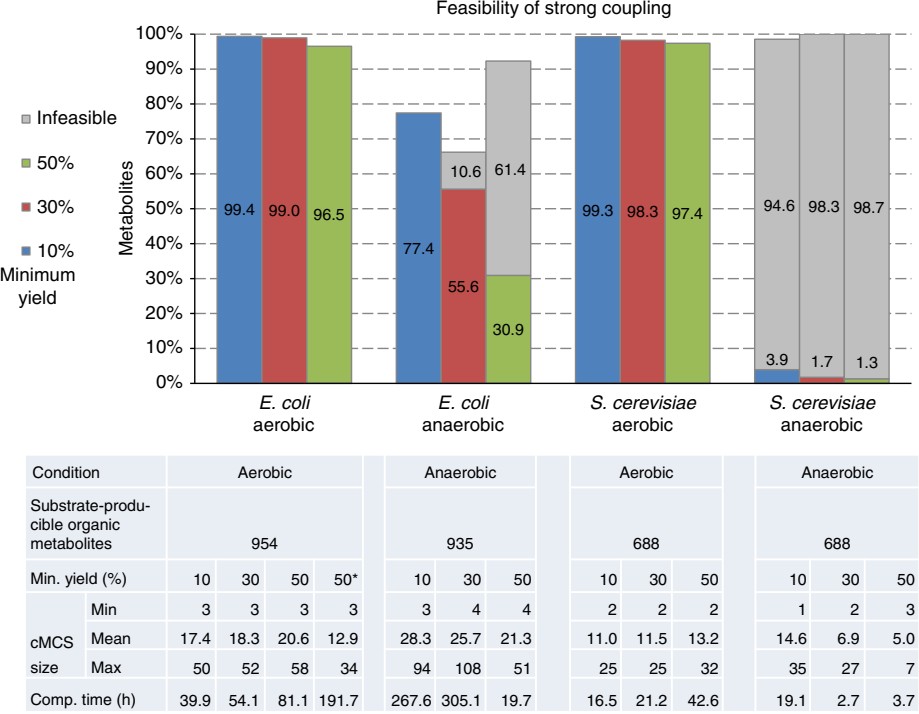

**Figure 2 | Feasibility of strong coupling in *Escherichia coli* and *Saccharomyces cerevisiae*.** Percentage of substrate-producible organic metabolites from *E. coli* and *S. cerevisiae* for which feasibility (coloured bars) or infeasibility (grey bars) of strong coupling can be proven, depending on the minimum yield level and conditions used. Statistics for the cut set sizes and MILP computation time are given in the table. The column (50*) for *E. coli* (aerobic growth) shows the cMCS sizes that result when the solver is restarted from the solution associated with the original cut set (for 50% minimum yield level) and minimization of the number of cuts is continued for up to 10 min for each cMCS. Sixty-eight of the cMCS found in this way can even be proven to be the smallest cMCS, that is, the solver has found an optimal solution.

even be seen as lower bounds. Figure 2 also shows the number of tested (substrate-producible) candidate metabolites in each model together with statistics about the size of the calculated cMCS and the computation time. As can be seen, the computation times in the aerobic scenarios increase with the demanded yield. This is in part due to the fact that when the (in-)feasibility of coupling cannot be decided for a given metabolite, then the associated MILP has been repeatedly executed a number of times until the time limit is reached, which increases the overall computation time considerably. For the cMCS sizes under aerobic conditions, a trend towards larger sizes with increasing minimum yield can be observed.

The mean cMCS sizes calculated for the organisms are partially quite large because genome-scale metabolic networks are used and, to keep the overall computation time acceptable, only limited time resources could be invested to minimize the size of each

cMCS (compare steps 7 and 10 in Methods section). To analyse whether smaller (and thus for practical applications more realistic) cMCS exist for the 50% minimum product yield level under aerobic conditions in *E. coli*, the MILP was restarted from the solution associated with the original cut set and minimization then continued for up to 10 min per metabolite (see column (50*) in Fig. 2). This required 8 days of additional computation time but reduced the mean cMCS size from 20.6 to 12.9 and the maximum cMCS size from 58 to 34 (the size histogram for these cMCS is shown Supplementary Data 1). Furthermore, 68 of the cMCS found during this extended time limit for minimization are already proven to be optimal, that is, they are the smallest cMCS (all of these contain at most six knockouts). This shows that for many cases cMCS with substantially reduced sizes can be found within a reasonable amount of time if efficient metabolic design strategies for coupling are to be calculated for specific products.

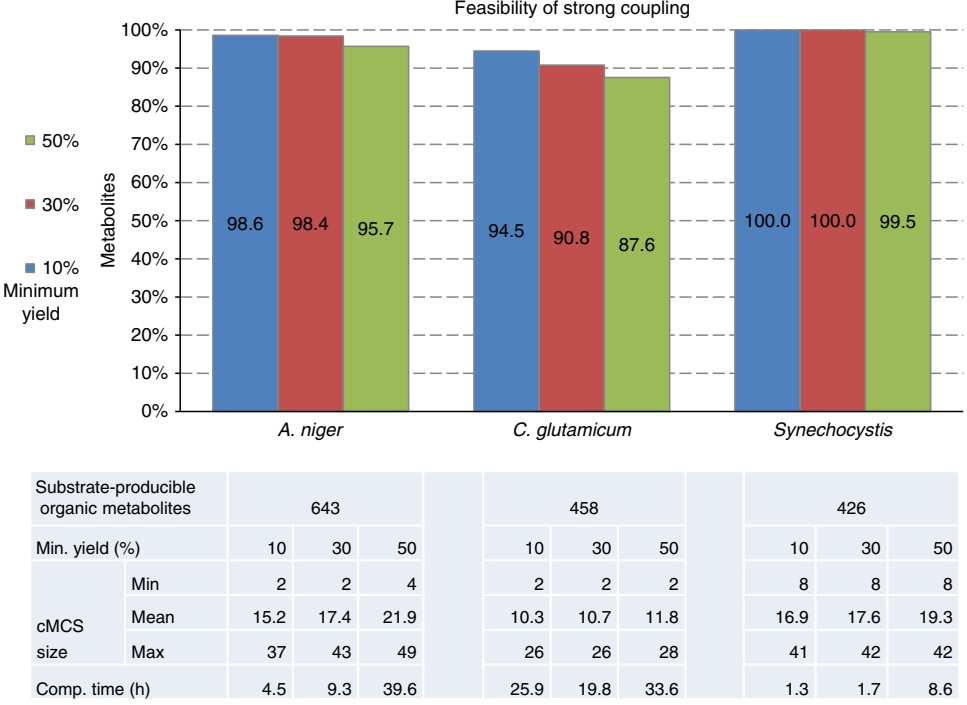

| Substrate-producible organic metabolites | | 643 | | | 458 | | | 426 | | |
|---|---|---|---|---|---|---|---|---|---|---|
| Min. yield (%) | | 10 | 30 | 50 | 10 | 30 | 50 | 10 | 30 | 50 |
| cMCS size | Min | 2 | 2 | 4 | 2 | 2 | 2 | 8 | 8 | 8 |
| | Mean | 15.2 | 17.4 | 21.9 | 10.3 | 10.7 | 11.8 | 16.9 | 17.6 | 19.3 |
| | Max | 37 | 43 | 49 | 26 | 26 | 28 | 41 | 42 | 42 |
| Comp. time (h) | | 4.5 | 9.3 | 39.6 | 25.9 | 19.8 | 33.6 | 1.3 | 1.7 | 8.6 |

**Figure 3 | Feasibility of strong coupling in *Aspergillus niger*, *Corynebacterium glutamicum* and Synechocystis species PCC 6803.** Percentage of substrate-producible organic metabolites from *A. niger*, *C. glutamicum* and *Synechocystis* species PCC 6803 for which feasibility of strong coupling can be proven, depending on the minimum yield level. For the practically obligate aerobes *A. niger* and *C. glutamicum* aerobic growth on glucose was considered while photoautotrophic metabolism was assumed for *Synechocystis*. Statistics for the cMCS sizes and MILP computation time are shown in the table.

For *E. coli* and yeast, we also analysed feasibility of strong coupling under anaerobic conditions. Here it should first be noted that, in our simulations, the degree of feasibility of coupling under anaerobic conditions can never be greater than for aerobic conditions since a deactivation of respiratory reactions in the cut sets for aerobic conditions can always mimic an anaerobic regime (in fact, some 'aerobic' cut sets target reactions involved in respiration). In *E. coli,* we found that the production of 77.4% of the metabolites producible from glucose under anaerobic conditions can still be coupled to growth at the 10% minimum yield level. With increasing demand for product yield, this fraction drops more strongly than in the aerobic case. Interestingly, because of the reduced solution space of flux vectors under anaerobic conditions, infeasibility of coupling can now be proven by the solver for a larger number of metabolites. For example, at the 50% minimum yield level, strong coupling was proven to be feasible for 30.9% and to be infeasible for 61.4% of the substrate-producible metabolites; hence, the percentage of feasible couplings is between 30.9 and 38.6%. The situation in yeast changes much more drastically when moving from an aerobic to an anaerobic growth regime since coupling becomes almost impossible for all metabolites. Even at the 10% minimum yield level, the fraction of substrate-producible metabolites that can be coupled drops to 3.9% and infeasibility can already be proven for more than 94%. We analysed in detail what structural properties of the yeast metabolism induce these sharp differences compared to *E. coli*. In contrast to the latter, in the chosen yeast model with standard outflows we found that excretion of ethanol is essential for anaerobic growth on glucose confirming experimental results[35]. After breakdown of glucose to glyceraldehyde 3-phosphate, ethanol is produced from glyceraldehyde 3-phosphate via 1,3-bisphospho-D-glycerate, 3-phospho-D-glycerate, 2-phospho-D-glycerate, phosphoenolpyruvate (PEP), pyruvate and acetaldehyde in a series of reaction steps that become essential under anaerobic conditions and are thus unavailable as knockout targets. In contrast, under anaerobic conditions in *E. coli* only the two reactions from glyceraldehyde 3-phosphate along 1,3-bisphospho-D-glycerate to 3-phospho-D-glycerate are essential. Since in yeast ethanol synthesis along the path listed above must be kept active in the model, strong coupling is lost for almost all metabolites: no suitable knockout sets can then exist that would guarantee a minimum yield of the respective metabolite because the substrate glucose could, in principle, be completely converted to ethanol. In fact, it has been reported that formation of ethanol as an undesired by-product is one disadvantage when establishing new fermentation processes based on yeast[36]. Only very few metabolites (for example, isobutanol, 2,3-butanediol) could, at least stoichiometrically, serve as alternative fermentation products in the model allowing disruption of pathways leading to ethanol, thus enabling strong coupling.

To illustrate how the fluxes in a metabolic network are affected by a cMCS, we describe the effects of the found cMCS inducing growth-coupled production of shikimate in yeast under aerobic conditions. The cMCS, which ensures a shikimate yield above 50% of its maximum yield, consists of three targets (compare Supplementary Data 1): {{PGCD [phosphoglycerate dehydrogenase] OR PSERT [phosphoserine transaminase] OR PSP_L [phosphoserine phosphatase (L-serine)]} AND {PYK [pyruvate kinase]} AND {TPI [triose-phosphate isomerase]}}. It tells us that for the first cut one of three reactions (phosphoglycerate dehydrogenase, phosphoserine transaminase, phosphoserine phosphatase) can be selected, which effectively serves to disrupt the phosphoserine pathway of serine biosynthesis, which connects to glycolysis via glyceraldehyde 3-phosphate. Since the second cut targets the triose-phosphate isomerase, glyceraldehyde 3-phosphate cannot be converted to

dihydroxyacetone phosphate. Therefore, glycolytic flux that flows through glyceraldehyde 3-phosphate has to proceed towards PEP, but cannot continue all the way to pyruvate because the pyruvate kinase is knocked out as third intervention. PEP, together with erythrose 4-phosphate from the pentose pathway, serves as entry point to the shikimate pathway and the further reaction steps towards shikimate all become essential. All in all, the cMCS channels an excess glycolytic flux towards PEP, which is then relieved via the production of shikimate. Since the cMCS contains the pyruvate kinase as knockout, it also becomes clear that this cMCS cannot work under anaerobic conditions because, as mentioned above, the pyruvate kinase is an essential reaction in yeast under anaerobic conditions.

Supplementary Data 1 also provides a list of the reactions in the *E. coli* model sorted with respect to the frequency of their occurrence in the found cMCS. As can be expected, reactions lying on pathways to standard (fermentation) products of *E. coli* (for example, lactate dehydrogenase, acetate kinase, acetaldehyde dehydrogenase) are frequently used targets.

The results of the computations for aerobic growth of *A. niger* and *C. glutamicum* on glucose and for photoautotrophic growth of *Synechocystis* sp. PCC 6803 lead to very similar findings as for aerobic growth in *E. coli* and yeast (Fig. 3). Growth-coupled designs can again be found for all three organisms for almost all metabolites. For example, for the 10% yield level, suitable intervention strategies exist in all three organisms for at least 94% of the metabolites and only a small reduction of this percentage (not below 87%) is seen for larger product yields. The highest percentage of feasibility of growth coupling for all investigated organisms can be seen for the phototrophic *Synechocystis* sp. PCC 6803, which can, at least partially, be attributed to the fact that this network model does not contain (irrepressible) exchange reactions for organic metabolites, which simplifies the induction of coupling.

The cMCS calculated for the five organisms directly target the reactions as they are contained in the models. However, although these reaction cut sets simplify the interpretation of the found intervention strategies as illustrated above, in reality the cuts must usually be implemented as gene knockouts. Owing to isozymes (encoded in different genes), enzyme complexes (whose parts are encoded in several genes) or multifunctional enzymes (which may catalyse more than one reaction), suitable cut sets with gene knockouts may differ from the reaction cut sets and one may ask whether the results on the feasibility of coupling holds also true with gene knockouts as relevant interventions. If the association between genes, enzymes and reactions is known and provided in the model, then the cMCS can also be calculated with gene knockouts by integrating the gene association into the model[37] (see Methods section). The *E. coli* iJO1366 (ref. 26) model contains well-established gene associations in the form of logical expressions for almost all reactions, which we used to check whether the feasibility of coupling is impacted when the cMCS are calculated as gene knockouts. In fact, in the aerobic case, the metabolites that can be coupled are nearly identical for all three minimum yield levels, thus confirming the broad feasibility of growth-coupled production. Only for one metabolite (protein-bound lipoate) the feasibility of coupling could not be decided when using gene knockouts, whereas a cMCS was found when using reaction cuts. For the anaerobic case the situation is more complicated: here the number of couplings found is slightly reduced (66.3%/50.2%/26.0% at the 10%/30%/50% minimum yield level) compared to reaction cuts (Fig. 2). For a few metabolites, we also found gene cMCS that induces coupling where a corresponding reaction cMCS could not be found.

## Discussion

The central goal of this study was to investigate systematically for five major production organisms frequently used in biotechnological applications how far suitable intervention strategies exist by which stoichiometric coupling of growth and synthesis of native metabolites can be achieved at genome scale. The results of our study are highly encouraging as they show that, under appropriate conditions, it is possible to strongly couple the production of the large majority of metabolites to growth for the organisms investigated here. Our work thus proves that growth-coupled product synthesis is indeed a widely usable design principle for metabolic engineering applicable to diverse organisms for enhancing the production of a large variety of metabolic products.

The presented exhaustive and genome-scale study on feasibility of growth-coupled strain designs is by far the largest and most comprehensive of its kind and our developed algorithmic pipeline turned out to be a very efficient and fast procedure for this purpose. So far, studies on feasibility of growth-coupled product synthesis focused on single products or/and on a single organism (*E. coli*) only[38,39]. Furthermore, although other used methods such as OptKnock[13], OptGene[40], GDLS[41] and FastPros[38] demand only weak coupling, which is easier to achieve than strong coupling demanded in this work, an almost unlimited feasibility of coupling under aerobic conditions in *E. coli* as proven herein could not be concluded with any of these methods[38].

For *E. coli* and especially for yeast the results show that, under anaerobic conditions, the feasibility of coupling drops markedly and infeasibility of coupling can be proven for a larger fraction of metabolites. In fact, coupling becomes even largely impossible in yeast. There are two possible reasons for this observation: First, during anaerobic growth it is necessary to remove excess NADH, which the cell can achieve by excreting fermentation products that are less oxidized than the substrate. This means that under anaerobic conditions outflow of some fermentation product(s) must be possible but must be restricted by reaction cuts in such a manner that not too much carbon is lost through fermentation because otherwise it will not be possible to keep up the minimum product yield. Therefore, coupling can be expected to be more difficult to realize (especially for yeast, where ethanol occurs as an essential by-product in the model) than under aerobic conditions where a possible NADH excess can be removed through respiration. This is related to the second possible reason why coupling is easier under aerobic conditions. Owing to respiration a much higher amount of ATP is available, which can support production of metabolites with high energy demand. However, for cases where no suitable knockout strategy for coupling could be found, it has to be noted that all calculations for the heterotrophic organisms were made based on glucose as substrate. We expect that, at least for some products, an infeasible coupling might become feasible with other substrates or if heterologous reactions or pathways are added.

The cMCS calculated here are primarily intended to test whether coupling is, in principle, possible or not. Existence of a suitable cMCS proves stoichiometric feasibility of coupling but does not consider regulatory (for example, feedback) or capacitive constraints (which might require further interventions or modifications; for example knockout of certain regulators, enzyme redesign or overexpression of certain genes). Furthermore, although measures have been taken to make sure that the cMCS are biologically plausible (knockouts of transport and non-enzymatic reactions were not allowed), not all calculated cMCS will represent suitable candidates for the construction of real production strains. When determining cMCS for experimental implementation, the time for minimization of the

cut sets should be extended (see below) and a variety of cMCS can be calculated from which some might be potentially more promising (for example, because of knowledge not contained in the model) than the others. In addition, higher minimum yields can be tested for production strains to determine what maximal product yields under coupling can be achieved with a limited number of knockouts.

The number of knockouts to be implemented is a relevant criterion for assessing the feasibility of a knockout strategy. For a smaller fraction of metabolites, even after spending more time for minimization, we identified very large cut sets. As an example, Supplementary Data 1 shows the histogram of the cut set sizes found for the extended computation for the 50% yield threshold in E. coli (where the average cut set size is 12.9; compare column 50* in Fig. 2). In all, 4.4% of the found intervention strategies would involve more than 20 reaction knockouts that might appear unrealistic. In those cases, as mentioned above, for a single (particular) product of interest, one may drastically increase the computation time to further reduce the cut set size, if possible all the way to the optimum. If the found cut sets are still (too) large, some of the targeted pathways, especially in the anabolism, can often be assumed to have a very low capacity and could therefore be excluded when implementing the knockout strategies, at least in a first attempt. Furthermore, given the ongoing evolution of genome-editing techniques, the experimental implementation also of cut sets with a larger number of knockouts can be expected to be feasible, especially in a model organism like E. coli. For instance, a well-known technique[42] for deleting arbitrary genes in E. coli has already been published in the year 2000, which requires about 6 days to establish a knockout. Recently, a similar technique has been proposed which enabled the implementation of seven gene knockouts in only 7 days[43]. Mutant strains with up to 16 reaction knockouts appear therefore not unrealistic anymore, with which more than 75% of the cut sets found in the extended E. coli 50% yield scenario would already become feasible. Finally, the CRISPR-Cas9 system has recently been shown to be a very efficient tool for multiple genetic manipulations, also in more complex organisms[44], and its particular potential for metabolic engineering has been emphasized[45].

An important factor when setting up the model for cMCS calculation for growth-coupled product synthesis is the selection of active organic metabolite outflows. In the model configurations used herein, we allowed standard (fermentation) products to be excreted by the cells. If outflows for other organic metabolites, that are unlikely to be excreted, are left open, then the number of required cuts will increase making the calculation and experimental implementation of the cut sets more difficult than necessary. Moreover, feasibility of coupling can then even be completely lost for some metabolites because no knockout strategy can be found that can prevent synthesis of undesired by-products while still allowing growth. For instance, when all 285 organic metabolite outflows contained in the iJO1366 model are open in E. coli, the percentage of the substrate-producible metabolites that can be coupled with growth at the 10% minimal product yield level reduces to 43.4%, which is nevertheless still significant. The mean cut set size increases by approximately seven reaction knockouts. Analogously, in yeast 52.4%, in A. niger even 81.4% and in C. glutamicum 35.9% of the metabolites can still be coupled at the 10% minimal yield level when all organic outflows present in the respective models are open (the Synechocystis model does not contain such outflows; see Methods section). In those cases, feasibility of coupling would increase again, if we allow knockouts also for at least those exchange reactions where corresponding genes of the involved transporters are definitely known (in fact, most transport reactions in the

iJO1366 (ref. 26) model have been assigned associated genes). Generally, opening all potential outflows, for example, in the E. coli model describes an extreme and unrealistic situation since normally no or only few organic metabolites (mainly standard fermentation products) are excreted by E. coli. On the other hand, if a production strain constructed from a cut set excretes a metabolite whose exchange reaction was not open in the model, then this cut set might not work as expected. For those cases, a practical solution can be found as follows: when the situation arises that, after experimental implementation of some knockouts of a calculated cut set, a metabolite is excreted whose outflow was not considered in the model before, it is possible to modify the model accordingly and then to recalculate and adapt the current cut set(s) to get intervention strategies which additionally suppress the unwanted excretion. In this manner, a production strain can be designed through an iterative cycle of calculation and experiment as was recently demonstrated for high-yield itaconic acid synthesis in E. coli[22].

In summary, our results underline the great potential of growth-coupled designs for the rational engineering of cell factories. We have shown that such designs are, in principle, widely realizable in all production organisms investigated. Several microbial strains that implement coupling have already been developed[8,10,12,17–23] and with our results we expect further reports of successful constructions of growth-coupled production strains in the future.

## Methods
**Model configurations.** The organisms and associated models used herein are listed in Table 1. The growth of the four heterotrophic organisms was simulated on glucose minimal medium. For E. coli and S. cerevisiae aerobic as well as anaerobic growth was considered. Simulation of anaerobic growth of E. coli was implemented by removing the exchange reaction for oxygen, while for S. cerevisiae this was achieved by removing the cytochrome c oxidase which is part of the respiratory chain (in the yeast model the production of a few essential metabolites requires oxygen, although in very small amounts only). The other two heterotrophic organisms, A. niger and C. glutamicum, are obligate aerobes; hence, only aerobic growth was considered. The fifth organism, Synechocystis sp. PCC 6803, is photoautotrophic and was simulated with light as limited 'substrate' together with an unlimited supply of $CO_2$. All models allow an unlimited uptake of inorganic compounds necessary for growth while uptake of the substrate (photons in case of Synechocystis sp. PCC 6803) is limited to known maximal values. Furthermore, organism-specific ATP requirements for non-growth-associated maintenance processes were taken into account if provided by the original models (Table 1).

The models of the four heterotrophic organisms contain many exchange reactions for organic metabolites allowing the outflow of the associated metabolites from the cell. For example, the E. coli model contains 285 and the yeast model 153 potential outflows for organic metabolites. As it is unlikely that all these organic metabolites are simultaneously excreted from the cell, the (non-essential) outflows of organic metabolites were restricted to typical (fermentation) products for the given organism when the cut sets were calculated (no restrictions were set for inorganic compounds). Concretely, in E. coli we considered ethanol, lactate, formate, acetate, succinate and hydrogen as possible outflows (methanol can also be excreted in the E. coli model, but occurs only in tiny amounts as by-product of biotin synthesis). For S. cerevisiae we allowed ethanol, glycerol, pyruvate, acetate and succinate to leave the cell. For A. niger the open outflows are gluconate, citrate, oxalate, malate, succinate and erythritol; for C. glutamicum they are glutamate, succinate, lysine, lactate, acetate, alanine, isoleucine and glycine. In the Synechocystis sp. PCC 6803 model only three organic metabolites can be excreted from the cell and because these outflows are essential for growth they were left open.

To ensure that the calculated knockout strategies (cut sets) have a high degree of biological relevance, reactions were set to be irrepressible in the models if they do not correspond to enzyme-catalysed biochemical reactions in which substrates are converted to products. This pertains to pseudo reactions representing transport processes, for example, between different cellular compartments and the exchange of substances to/from the extracellular space (even if genetically encoded transporters are known the corresponding reactions were considered to be not repressible as these transporters are often unspecific). Furthermore, other pseudo reactions (including the consumption of ATP for maintenance processes) and non-enzymatic spontaneous reactions are contained in the models. All these reactions mentioned above were considered as irrepressible, that is, they cannot be knocked out in the cut sets (Table 1). In the E. coli model, all reactions that do not have an associated gene were also considered irrepressible. For the other two models that

include some gene-reaction mappings (*A. niger* and *C. glutamicum*), the cases of missing associations were investigated in more detail to decide whether or not to make such reactions irrepressible. The number and percentage of irrepressible reactions in the respective models is shown in Table 1.

**Procedure for checking the feasibility of growth-coupled synthesis.** Each metabolic network with $m$ internal metabolites and $n$ reactions is represented by its $m \times n$ stoichiometric matrix $\mathbf{N}$ together with the sets *Irr* and *Rev* containing the indices of the irreversible and reversible reactions, respectively. The network is assumed to be in steady state implying that the net reaction rates $\mathbf{r} = (r_1, r_2, \ldots, r_n)^{\mathrm{T}}$ fulfil

$$\mathbf{Nr} = \mathbf{0}, r_i \geq 0 \; \forall i \in Irr.$$

For some reactions, lower ($\alpha_i$) or/and upper ($\beta_i$) flux bounds might be known further constraining the reaction rates:

$$a_i \leq r_i \leq \beta_i.$$

Metabolic flux distributions in a cell that are unfavourable for the efficient production of a certain chemical (for example, flux vectors with low product yield $Y^{\mathrm{P/S}}$) can be specified by linear inequalities

$$\mathbf{Tr} \leq \mathbf{t}$$

with $t \times n$ matrix $\mathbf{T}$ and $t \times 1$ vector $\mathbf{t}$. Herein, we used the following inequality to describe undesired flux distributions having a product yield below a given minimum threshold $Y^{\mathrm{P/S}}_{\min}$ ($r_{\mathrm{S}}$ is the substrate uptake rate and $r_{\mathrm{P}}$ the product excretion rate):

$$\frac{r_{\mathrm{P}}}{r_{\mathrm{S}}} \leq Y^{\mathrm{P/S}}_{\min} \Leftrightarrow r_{\mathrm{P}} - Y^{\mathrm{P/S}}_{\min} \cdot r_{\mathrm{S}} \leq 0 \qquad (1)$$

Hence, matrix $\mathbf{T}$ consists here of a single row containing zeros except a '$+1$' for $r_{\mathrm{P}}$ and $-Y^{\mathrm{P/S}}_{\min}$ in the column of $r_{\mathrm{S}}$, while vector $\mathbf{t}$ has only one (row) element being zero.

Similarly, the inequalities

$$\mathbf{Dr} \leq \mathbf{d}$$

with $d \times n$ matrix $\mathbf{D}$ and $d \times 1$ vector $\mathbf{d}$ can be used to represent desired (wanted) metabolic behaviours. We used the following inequalities to describe desired flux distributions with a product yield above $Y^{\mathrm{P/S}}_{\min}$ together with a minimum biomass yield $Y^{\mathrm{B/S}}_{\min}$ ($\mu$ is the growth rate):

$$\begin{aligned} Y^{\mathrm{P/S}}_{\min} \cdot r_{\mathrm{S}} - r_{\mathrm{P}} \leq 0 \\ Y^{\mathrm{B/S}}_{\min} \cdot r_{\mathrm{S}} - \mu \leq 0 \end{aligned} \qquad (2)$$

Hence, matrix $\mathbf{D}$ consists here of two rows: the first contains zeros except a '$-1$' for $r_{\mathrm{P}}$ and $Y^{\mathrm{P/S}}_{\min}$ in the column of $r_{\mathrm{S}}$, while the second row contains non-zero values only for the growth rate ($-1$) and again for the substrate uptake rate $r_{\mathrm{S}}$ (value $Y^{\mathrm{B/S}}_{\min}$). The vector $\mathbf{d}$ has two rows both containing a zero.

For inducing (and proving feasibility of) strong coupling, a reaction knockout set (a cMCS) has to be found that disables all undesired flux vectors (fulfilling (1)) while keeping at least one desired flux vector fulfilling (2). Note that the desired behaviour described in (2) is a subset of the complement of the undesired behaviour in (1) as the latter does not contain a constraint for biomass yield.

Given these specifications, the MILP problem that is used to calculate a cMCS to check if growth-coupled production is possible takes the following form[34,46]:

$$\begin{pmatrix} \mathbf{N}_{\mathrm{Irr}}^{\mathrm{T}} & \mathbf{I}_{\mathrm{Irr}} & \mathbf{0} & \mathbf{0} & \mathbf{T}_{\mathrm{Irr}}^{\mathrm{T}} & \mathbf{0} \\ \mathbf{N}_{\mathrm{Rev}}^{\mathrm{T}} & \mathbf{0} & \mathbf{I}_{\mathrm{Rev}} & -\mathbf{I}_{\mathrm{Rev}} & \mathbf{T}_{\mathrm{Rev}}^{\mathrm{T}} & \mathbf{0} \\ \mathbf{0} & \mathbf{0} & \mathbf{0} & \mathbf{0} & \mathbf{0} & \mathbf{N} \\ \mathbf{0} & \mathbf{0} & \mathbf{0} & \mathbf{0} & \mathbf{0} & \mathbf{D} \end{pmatrix} \begin{pmatrix} \mathbf{u} \\ \mathbf{vp}_{\mathrm{Irr}} \\ \mathbf{vp}_{\mathrm{Rev}} \\ \mathbf{vn}_{\mathrm{Rev}} \\ \mathbf{w} \\ \mathbf{r} \end{pmatrix} \begin{matrix} \geq \\ = \\ = \\ \leq \end{matrix} \begin{pmatrix} \mathbf{0} \\ \mathbf{0} \\ \mathbf{0} \\ \mathbf{d} \end{pmatrix}$$

$$\mathbf{t}^{\mathrm{T}}\mathbf{w} \leq -c$$

$\mathbf{u} \in \Re^m, \; \mathbf{r} \in \Re^n, \; \mathbf{w} \in \Re^t, \; \mathbf{vp}_{\mathrm{Irr}}, \mathbf{vp}_{\mathrm{Rev}}, \mathbf{vn}_{\mathrm{Rev}}, \mathbf{w}, \mathbf{r}_{\mathrm{Irr}} \geq \mathbf{0}, \; c > 0$

$\forall i \in Rev : r_i \geq (1 - zp_i - zn_i) \cdot \alpha_i, \; r_i \leq (1 - zp_i - zn_i) \cdot \beta_i, \; zp_i + zn_i \leq 1$

$\forall i \in Irr : r_i \geq (1 - zp_i) \cdot \alpha_i, \; r_i \leq (1 - zp_i) \cdot \beta_i$

$zp_i, zn_i \in \{0, 1\}, \; \alpha, \beta \in \Re^n$

Here the stoichiometric matrix $\mathbf{N}$, the identity matrix $\mathbf{I}$ and the matrix $\mathbf{T}$ are split into two submatrices containing the reversible ($\mathbf{N}_{\mathrm{Rev}}$, $\mathbf{I}_{\mathrm{Rev}}$ and $\mathbf{T}_{\mathrm{Rev}}$) and irreversible ($\mathbf{N}_{\mathrm{Irr}}$, $\mathbf{I}_{\mathrm{Irr}}$ and $\mathbf{T}_{\mathrm{Irr}}$) reactions (columns), respectively. The $zp_i$ and $zn_i$ variables are Boolean indicator variables that distinguish whether the corresponding $vp_i$ and $vn_i$ variables are equal or unequal to zero ($zp_i = 0 \leftrightarrow vp_i = 0$, $zp_i = 1 \leftrightarrow vp_i \neq 0$ for all reactions and additionally $zn_i = 0 \leftrightarrow vn_i = 0$, $zn_i = 1 \leftrightarrow vn_i \neq 0$ for the reversible reactions). In case an indicator variable is unequal to zero, then its associated reaction is in the cut set and can carry no flux as demanded by the constraints for $r_i$. For this MILP problem it is essential that finite lower ($\alpha_i$) and upper ($\beta_i$) bounds for all fluxes are provided (see below). Finally, for the irrepressible reactions that are not allowed to be knocked out (see above and Table 1), the values of their associated $zp_i$ and, in case of reversible reactions, of $zn_i$ variables are fixed to zero in the MILP.

The MILP explained above is the central element of the procedure for testing whether a suitable knockout strategy (cMCS) exists that induces growth-coupled production of a metabolite with a demanded minimum product yield. In each model, the feasibility of growth coupling is checked for all organic metabolites that can be produced from the substrate, with two exceptions: First, for the models that use glucose as substrate the possibility of coupling the production of unphosphorylated glucose (which may occur in other compartments beside the extracellular space) is not considered as it is the same compound as the substrate. Second, if a metabolite (for example, ethanol or acetate in the *E. coli* and yeast model) is connected to one of the open standard outflow pathways of the model (which may go through different compartments), then it is assumed that the metabolite to be coupled is excreted along this outflow. Hence, in those cases, coupling is only considered for the (excreted) metabolite in the extracellular space and different instances of the same metabolite in other compartments are not taken as candidates for coupling. The respective numbers of candidate metabolites for coupling are shown in Figs 2 and 3 ('substrate-producible organic metabolites') and full lists of the metabolites that are candidates for coupling can be found in Supplementary Data 1.

For each candidate metabolite for coupling, the following steps are performed:

(1) An exchange reaction for this metabolite is added to the model if it does not yet exist. The exchange reaction is set up so that only the outflow (not uptake) of the metabolite is possible.

(2) The flux through this export reaction is maximized via solving an appropriate linear optimization (LP) problem (applying substrate uptake limit, ATP maintenance). If the result is zero, then the metabolite cannot be produced at all, if its unbounded then its production is not bounded by the limited substrate (those metabolites are not part of the list of candidate metabolites for coupling). When the result is greater than zero and bounded it is divided by the substrate uptake and taken as maximum product yield.

(3) The minimum demanded product yield $Y^{\mathrm{P/S}}_{\min}$ is set to the required fraction (10%/30%/50%) of the maximum product yield; the minimum demanded biomass yield $Y^{\mathrm{B/S}}_{\min}$ is set to 0.01 gDW mmol$^{-1}$ glucose for the heterotrophic organisms and to $10^{-4}$ gDW mmol$^{-1}$ photons for the cyanobacterium *Synechocystis* sp. PCC 6803 (which requires 51 photons to produce one molecule glucose).

(4) The network is compressed by merging sets of fully coupled reactions and by removing conservation relations.

(5) In case a cut set for a lower minimum product yield is known, it is checked whether this cut is also applicable for the current minimum yield. If this is the case steps 6 to 8 are skipped.

(6) A flux variability analysis (FVA) with substrate uptake limit, ATP maintenance and minimum biomass yield $Y^{\mathrm{B/S}}_{\min}$ is performed to calculate flux bounds for all reactions; in case unbounded fluxes remain these are limited to $\pm 2,000$ mmol gDW$^{-1}$ h$^{-1}$. Compared to the LP in step 2, minimum biomass yield has been added as additional restriction for the FVA. Consequently, the LPs of the FVA may now be infeasible in which case growth-coupled production is not possible for this metabolite (where the following steps can be skipped).

(7) The MILP is run with a given time limit (1–10 min.). The MILP minimizes the number of knockouts, but, to reduce the computation time, the solver is configured to stop as soon as the relative gap between the current objective and the best bound drops below 98% (which is still large but sufficient for our purpose). The solver stops when a solution is found, the time limit is reached or when the problem is determined by the solver to be infeasible (meaning that coupling is not possible).

(8) In case a solution (and thus a cut set) has been found by the solver for the MILP problem it is verified with separate LPs, testing whether, under application of the reaction knockouts contained in the cut set, coupling is achieved, that is, the undesired behaviour becomes infeasible and the desired behaviour remains feasible.

(9) If neither a solution was found nor the infeasibility of the problem has been determined, then the procedure is repeated up to 10 times from step 7 using a different solver seed which leads to a different exploration of the search space.

(10) If a cut set has been found this will typically be a *non-minimal* cut set; hence, a superset of one or more cMCS. A cMCS is then extracted from the cut set by iteratively checking the necessity of each knockout with a LP (this yields a cMCS since no further knockout can be removed from the cut set; however, it is not necessarily the smallest cMCS with fewest number of cuts).

When no cMCS was found by the solver nor the infeasibility of the problem concluded after the maximum number of repetitions of steps 7 and 8, then it cannot be decided whether growth coupling is possible or not. If, for a given product and coupling yield (for example, 30%), a cut set $C_1$ was found that required less interventions than a cut set $C_2$ found for the same metabolite for a lower coupling yield (for example, 10%), then cut set $C_2$ was replaced by $C_1$ (this is mainly relevant for the cut set size statistics shown in Figs 2 and 3).

For the calculation of gene cut sets in the *E. coli* model, we adapted a recently proposed approach[37] and integrated the gene–enzyme–reaction association into the metabolic network model as follows: for each enzyme-catalysed reaction an

auxiliary metabolite is added which is consumed by this reaction. Each auxiliary metabolite is produced by one or more reactions with each of these reactions corresponding to an enzyme that can catalyse the metabolic reaction associated with the auxiliary metabolite. A reaction that corresponds to an enzyme thereby consumes metabolites which represent the gene product(s) of which this enzyme is composed. Each gene product metabolite is in turn produced by a gene translation reaction (which does not consume anything). To calculate gene cuts only the gene translation reactions are allowed to be knocked out. Furthermore, only those gene translation reactions can be cut that affect metabolic reactions which are repressible in the reaction cut calculations. All other reactions are considered to be irrepressible.

All calculations were carried out on a computer with two Intel Xeon X5650 (2.67 GHz) hexacore CPUs using API functions of CellNetAnalyzer[47] (version 2016.1) which uses CPLEX 12.5.1 as MILP solver.

**Code availability.** CellNetAnalyzer can be downloaded at: http://www2.mpi-magdeburg.mpg.de/projects/cna/cna.html. An example script to calculate cMCS for growth-coupled product synthesis can be found at: http://www2.mpi-magde-burg.mpg.de/projects/cna/etcdownloads.html.

**Data availability.** The authors declare that the data supporting the findings of this study are available within the paper and its Supplementary Data file.

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

## Acknowledgements

We are grateful to O. Hädicke for valuable comments. This work was in part supported by the German Federal Ministry of Education and Research (de.NBI partner project 'NBI-ModSim' (FKZ: 031L104B) and by the European Research Council (ERC Consolidator Grant 721176).

## Author contributions

S.K. conceived the study. A.v.K. implemented algorithms and performed the calculations. A.v.K. and S.K. analysed and discussed the results and wrote the manuscript.

## Additional information

**Competing interests:** The authors declare no competing financial interests.

