## [Peer review file · Nature Communications]

Reviewers' comments:

Reviewer #1 (Remarks to the Author):

In the current work the authors study the feasibility of product formation and growth coupling in 5 industrially relevant model organisms using native metabolites as target products.

The manuscript was clearly written. The manuscript presented interesting results regarding the existence of designs based on the previously conceived cMCS method. However, such designs do not imply that the product can be coupled in reality, as the authors acknowledge (line 302), regulatory and capacitive constraints are not accounted for. Indeed, the metabolic cost of expressing functional proteins that maintain flux through the production pathways, may result in a strain that does not meet the design criteria, such as minimum growth rate. Since the infeasibility of coupling based on metabolic reactions is conclusive, the results can be considered as an "upper bound" for the number of products that may be coupled metabolically. Overall, while the findings of the paper are interesting, the novelty is not remarkable, since it has not truly addressed the challenge of feasibility in a new way. Previous studies (Feist et. al, Metab Eng, 2010) have also applied established strain design algorithms to a plethora of products in order to study the viability of growth coupled production phenotypes. As a secondary suggestion, it would also be interesting to expand the list of metabolites considered by the authors. As many interesting metabolites are non-native to the presented organisms.

Reviewer #2 (Remarks to the Author):

The authors present here a computational study showing that growth coupling (i.e. coupling such that growth requires product formation) is feasible (aerobically) for a large fraction of metabolites in five different organisms. This result is very interesting, surprising and (potentially) has important practically relevant applications. This is particularly true given the availability of tools for Adaptive Lab Evolution and efficient genome engineering through CRISPR-cas9 (the coupling of this method with these tools should be mentioned in the manuscript, btw, in order to increase its appeal to the metabolic engineering field). The mathematical techniques are solid and have been tested before. I therefore explicitly recommend the publication of this manuscript.

However, before doing so I would require several modifications:

1-) The result is surprising, and many readers would have a hard time convincing themselves of the validity of the results when explained in such abstract terms. I would require that the authors choose a metabolite at random and explain in detail for that example why the growth coupling is possible. It would be best to do this for *S. cerevisiae*: pick one metabolite for which growth coupling is feasible (aerobically) and another one for which it is unfeasible (anaerobically).

2-) In lines 233-235 (page 8) it is mentioned that ethanol synthesis must be kept active for anaerobic growth. The reason for this difference with respect to *E. coli* should be

explained in terms of stoichiometry.

3-) Given the surprising results, the code should be made available in order to replicate them.

4-) In the caption of Figure 3 the reason for not considering the anaerobic case (i.e. obligate aerobes) should be explicitly mentioned.

5-) The nature of matrices T and D and vectors t and d in lines 420 (page 14) and 429 (page 15) should be explained so as to increase intelligibility.

Reviewer #3 (Remarks to the Author):

In their work, von Kamp and Klamt investigate the feasibility of coupling the production of certain metabolites of interest to growth (termed weak coupling) and substrate uptake (strong coupling) in five common production organisms. They show that for the conditions investigated, coupling can be achieved for most metabolites. While the underlying question is of particular relevance in the context of metabolic engineering, the manuscript has several important shortcomings.

1) The metabolic networks were modified such that only the excretion of typical fermentation products was allowed. While this might be reasonable (to a limited extent) in the context of wild-type growth it is likely that genetic perturbations lead to the excretion of further compounds. For instance, it has been found that knockouts in central metabolic pathways can result in an increased excretion of amino acids (Pande et al., ISME J. 2014, 8(5):953-62). Also for wild-type E. coli, it is known that some environmental conditions can lead to the excretion of particular products such as agmatine in acid stress or citrate during iron deprivation. Thus, computations should be performed with all admissible outflows (which the authors have apparently done as a test case for E. coli).

2) The constrained minimal cut sets (cMCSs) discussed in the paper are not necessarily the smallest ones. Indeed, the authors write that "only very limited computational resources were invested". However, the smallest cMCSs are of particular importance since, to determine whether it is indeed experimentally possible to implement such knockouts by genetic modifications, the number of necessary knockouts is a limiting factor. Moreover, the authors state that they used just 12 cores for their computations. Clearly, more computational resources should be available either locally, at collaboration partners or through national super computing centers. In my opinion clearly only smallest cMCSs should be provided and not intermediate solutions like it is the case in the present manuscript.

3) Beyond the discussion of flux coupling, there is only little biological insight provided in the manuscript while it could clearly benefit from a more in depth discussion. Question of interest could be, for instance, whether there is a certain pattern of reactions that are typically knocked out ("knock out hubs") or even specific metabolic subsystems that are frequently part of knockout strategies. This could also be brought into context with specific types of metabolites to be produced (e.g. alcohols, lipids, etc). This type of information should be easily available from the results already obtained and would considerably improve the content of the manuscript.

4) There is no discussion as to the experimental feasibility of the discussed knockout strategies since they are the time-limiting factor in strain implementation. In my opinion strategies up to six knockouts are probably practically feasible, but there should be an in depth discussion which number of knockouts is reasonable and, given that number, for which number of metabolites weak or strong coupling could still be achieved.

5) For yeast in the anaerobic case, coupling can only be achieved for a minority of metabolites. The authors state that this might be due to the mandatory ethanol excretion under these conditions that limit yield. I think in such a case the analysis could also be performed while excluding ethanol production in yield calculations to get a more realistic picture of the possibility of growth coupling in this condition.

Minor issues:

cMCS are defined on the level of reactions, while genetic implementation requires the knockout of individual genes. It should at least be discussed and exemplified for some cases whether determining cMCSs directly on the gene level would considerably change the conclusions drawn in the paper. For instance, one could test whether growth coupling persists when translating knocked out reactions into the corresponding gene sets and whether those gene sets are still minimal.

Response to reviewer comments

Reviewer #1 (Remarks to the Author):

In the current work the authors study the feasibility of product formation and growth coupling in 5 industrially relevant model organisms using native metabolites as target products.

The manuscript was clearly written. The manuscript presented interesting results regarding the existence of designs based on the previously conceived cMCS method.

We thank the reviewer for the positive assessment of the manuscript.

However, such designs do not imply that the product can be coupled in reality, as the authors acknowledge (line 302), regulatory and capacitive constraints are not accounted for. Indeed, the metabolic cost of expressing functional proteins that maintain flux through the production pathways, may result in a strain that does not meet the design criteria, such as minimum growth rate. Since the infeasibility of coupling based of metabolic reactions is conclusive, the results can be considered as an “upper bound” for the number of products that may be coupled metabolically. Overall, while the findings of the paper are interesting, the novelty is not remarkable, since it has not truly addressed the challenge of feasibility in a new way. Previous studies (Feist et. al, Metab Eng, 2010) have also applied established strain design algorithms to a plethora of products in order to study the viability of growth coupled production phenotypes.

We agree that the work of Feist et al. (Metab Eng, 2010) was an important milestone to study feasibility of growth-coupled production processes and we therefore cite this reference as a relevant and substantial previous study. At the same time we believe that our work goes far beyond the results presented in the cited paper. While Feist et al. sought to identify growth-coupled strain designs for 11 products in a *E. coli* genome-scale model, we searched for growth-coupled strain designs for 954 metabolites in *E. coli* and did the same for four other frequently used production organisms (hence, feasibility of coupling has been studied for 3111 metabolites in total). Moreover, although only weak coupling was demanded in the work by Feist et al., feasibility of coupling could be shown for 8 of the 11 metabolites whereas we found that even strong coupling is feasible for all these 11 metabolites. This discrepancy is most likely due to the fact that the number of knockouts was restricted to 10 in the original paper of Feist et al., whereas our computational framework is able to deal with practically arbitrary many KO's to demonstrate principal feasibility of coupling.

Generally, and most importantly, an almost unlimited feasibility of coupling as proven herein for *E. coli* (and other organisms) has not been concluded with any other method or in any other study before.

As a secondary suggestion, it would also be interesting to expand the list of metabolites considered by the authors. As many interesting metabolites are non-native to the presented organisms.

The focus of this contribution is on the native metabolism of the five production organisms. As the reviewer said, in principle, one could extend the analysis also to non-native metabolic products. However, to conduct this kind of screening rigorously and comprehensively, one would need to consider a huge master network of all known metabolic pathways and reactions; this would be the metabolism of a “super organism” where the metabolism of the considered host organism is contained as a sub-network. (Note that just introducing a heterologous pathway leading to a certain non-native product is, in general, not sufficient since other heterologous reactions might be required to achieve coupling of growth and product synthesis). Such a well-curated master network including all relevant heterologous pathways does not exist yet and its reconstruction would be a big challenge on its own. Generally, in our opinion, the analysis of non-native metabolites would go beyond the scope of our contribution. Since we have analyzed 5 different organisms including prokaryotes and eukaryotes as well as heterotrophic and photoautotrophic organisms, we believe that our study already proved feasibility of coupling for a very large number of metabolites and potential products.

Reviewer #2 (Remarks to the Author):

The authors present here a computational study showing that growth coupling (i.e. coupling such that growth requires product formation) is feasible (aerobically) for a large fraction of metabolites in five different organisms. This result is very interesting, surprising and (potentially) has important practically relevant applications.

We thank the reviewer for the very positive assessment of the manuscript.

This is particularly true given the availability of tools for Adaptive Lab Evolution and efficient genome engineering through CRISPR-cas9 (the coupling of this method with these tools should be mentioned in the manuscript, btw, in order to increase its appeal to the metabolic engineering field).

We agree that we should mention the importance of genome editing techniques such as CRISPR-Cas9 for the realization of intervention strategies (possibly requiring a larger number of gene knockouts) for growth-coupled product synthesis. We therefore included a respective paragraph in the Discussion section.

The mathematical techniques are solid and have been tested before. I therefore explicitly recommend the publication of this manuscript.

However, before doing so I would require several modifications:

*1-) The result is surprising, and many readers would have a hard time convincing themselves of the validity of the results when explained in such abstract terms. I would require that the authors choose a metabolite at random and explain in detail for that example why the growth coupling is possible. It would be best to do this for *S. cerevisiae*: pick one metabolite for which growth coupling is feasible (aerobically) and another one for which it is unfeasible (anaerobically).*

As an example, we now describe the effect of the cut set (cMCS) found for the growth-coupled

production of shikimate in yeast under aerobic conditions in the Results section and explain why this cMCS will not work under anaerobic conditions.

2-) *In lines 233-235 (page 8) it is mentioned that ethanol synthesis must be kept active for anaerobic growth. The reason for this difference with respect to E. coli should be explained in terms of stoichiometry.*

We have added a description of the reaction essentialities elicited by anaerobic conditions in E.coli and yeast to further elucidate the differences between these two metabolic network models.

3-) *Given the surprising results, the code should be made available in order to replicate them.*

As now also mentioned in the manuscript, an example script for the E. coli model illustrating how the results can be calculated using the CellNetAnalyzer API has been made available on <http://www2.mpi-magdeburg.mpg.de/projects/cna/etcdownloads.html>.

However, note that an exact replication of the cMCS reported here can not be expected because the results of time-constrained heuristic optimizations will vary depending on the CPLEX version and hardware. The percentage of metabolites for which coupling can be proven or ruled out should however be very similar as long as a comparable amount of computational effort is used. Furthermore, note that Supplemental Table 1 contains all found cut sets for growth-coupled production of metabolites (for all species/growth conditions). These cut sets can be easily tested for correctness by interested readers with the respective (publically available) models.

4-) *In the caption of Figure 3 the reason for not considering the anaerobic case (i.e. obligate aerobes) should be explicitly mentioned.*

We now mention in the Figure caption that A. niger and C. glutamicum are practically obligate aerobes.

5-) *The nature of matrices T and D and vectors t and d in lines 420 (page 14) and 429 (page 15) should be explained so as to increase intelligibility.*

We agree and added text explaining the shape of these matrices and vectors in our concrete setting.

Reviewer #3 (Remarks to the Author):

In their work, von Kamp and Klamt investigate the feasibility of coupling the production of certain metabolites of interest to growth (termed weak coupling) and substrate uptake (strong coupling) in five common production organisms. They show that for the conditions investigated, coupling can be achieved for most metabolites. While the underlying question is of particular relevance in the context of metabolic engineering, the manuscript has several important shortcomings.

1) *The metabolic networks were modified such that only the excretion of typical fermentation products was allowed. While this might be reasonable (to a limited extent) in the context of wild-type growth it is likely that genetic perturbations lead to the excretion of further compounds. For instance, it has been found that knockouts in central metabolic pathways can result in an increased excretion of amino acids (Pande et al., ISME J. 2014, 8(5):953-62). Also for wild-type E. coli, it is known that some environmental conditions can lead to the excretion of particular products such as agmatine in acid stress or citrate during iron deprivation. Thus, computations should be performed with all admissible outflows (which the authors have apparently done as a test case for E. coli).*

We agree that genetic perturbations may lead to the excretion of further compounds beyond the typical fermentation products. The problem is that it is not known beforehand which perturbations may lead to which excretions, but it appears to be quite unlikely (much more unlikely than the here assumed possible excretion of standard products) that such perturbations will lead to a situation where all organic metabolites, as far as their outflows are included in the model, are excreted by the organism. This would then require to block synthesis (pathways) of all those (285 !!) excretion metabolites which renders the size of the cut sets unnecessarily large. Therefore, as presented in the discussion, we see the situation where all organic outflows are open as a kind of worst-case scenario. We have performed the calculations for this scenario (now also for the other organisms) and mention the results in the discussion showing that the percentage of feasibility of growth-coupling is still unexpectedly large. Also, the discussion describes a possible strategy how to deal with the situation when the experimental implementation of knockouts leads to the unwanted excretion of additional metabolites.

2) *The constrained minimal cut sets (cMCSs) discussed in the paper are not necessarily the smallest ones. Indeed, the authors write that “only very limited computational resources were invested”. However, the smallest cMCSs are of particular importance since, to determine whether it is indeed experimentally possible to implement such knockouts by genetic modifications, the number of necessary knockouts is a limiting factor. Moreover, the authors state that they used just 12 cores for their computations. Clearly, more computational resources should be available either locally, at collaboration partners or through national super computing centers. In my opinion clearly only smallest cMCSs should be provided and not intermediate solutions like it is the case in the present manuscript.*

We agree that for experimental implementation it is desirable to calculate the smallest minimal cut sets (MCSs). However, the calculation of the guaranteed smallest MCS can be, computationally, extremely expensive because optimizing a MILP is a hard problem (NP complete). For problems of this type, heuristics have been developed that allow the comparatively fast calculation of suboptimal solutions and such heuristics are naturally employed by the CPLEX solver we use. However, this still means that, in general, it remains hard to determine the guaranteed optimal solution. In our experience, the time needed to prove the optimality of a solution appears to increase exponentially with the size of the smallest MCS. This is already observable in our results where we let the solver run for 10 further minutes after the initial solution was found because only for MCSs with a size up to six optimality could eventually be proven within this time limit. For problems with smallest MCSs comprising more than 10 knockouts we expect, at least partially, forbiddingly high computation times and memory requirements to find the smallest MCSs because the search space will be huge and the solver needs to prove that all smaller cut sets are infeasible. We therefore expect that, even on large clusters, the calculation of the smallest MCSs for products where the cardinality of these MCSs is large might be prohibitively high or even infeasible. It

also has to be taken into account that the multi-threaded optimization time for a MILP does not decrease linearly with the number of cores which means that using twice the number of cores will reduce the optimization time by significantly less than 50%. Furthermore, end-user support for distributed parallel MILP optimization has only recently been added to CPLEX and probably scales even less well than multi-threading. Therefore, the efficient use of increased computational power for solving a MILP is a non-trivial topic in itself which is outside of the scope of the current manuscript. Most importantly, since, in this work, we are primarily interested in the *principal feasibility* of growth-coupled product synthesis we consequently think that, at this stage, it is not necessary to spend more computational resources to look for the smallest MCS for all metabolites in all species. Of course, later, in concrete applications where the production of one particular metabolite is of interest, one may spend much more time to find (also a larger number of) the smallest MCSs for this single product. Nonetheless, the efficient use of computational resources is a topic for the further development of our cMCS algorithm.

Regarding the feasibility of the calculated cut sets with respect to their size, one should also note that the average size of the optimized cut sets in *E. coli*, is, even for the 50% product yield level (aerobic growth; see column 50* in Figure 2), approximately 13 thus indicating that for a large number of products realistic intervention strategies exist that can be implemented with modern genetic techniques (CRISPR etc.).

3) Beyond the discussion of flux coupling, there is only little biological insight provided in the manuscript while it could clearly benefit from a more in depth discussion. Question of interest could be, for instance, whether there is a certain pattern of reactions that are typically knocked out (“knock out hubs”) or even specific metabolic subsystems that are frequently part of knockout strategies. This could also be brought into context with specific types of metabolites to be produced (e.g. alcohols, lipids, etc). This type of information should be easily available from the results already obtained and would considerably improve the content of the manuscript.

The cMCS calculated here are exemplary cut sets that mainly serve to prove the feasibility of coupling. For the proper analysis of knockout hubs or similar network features we believe that a large variety of cut sets per metabolite or metabolite class should be sampled because when basing the analysis only on the exemplary cut sets calculated here it will quite likely be skewed.

To give at least an impression of what reactions are the most frequent targets, we included a list of reaction cut counts for *E. coli* in the Supplementary Table, which contains a classification of reactions into metabolic subsystems. However, we do not discuss these results in detail for the reasons mentioned above.

4) There is no discussion as to the experimental feasibility of the discussed knockout strategies since they are the time-limiting factor in strain implementation. In my opinion strategies up to six knockouts are probably practically feasible, but there should be an in depth discussion which number of knockouts is reasonable and, given that number, for which number of metabolites weak or strong coupling could still be achieved.

We have added a paragraph in the discussion about the implementation of knockouts and the limitation regarding large cMCS. As was also suggested by Reviewer 1, we also point to new genome editing techniques such as CRISPR-Cas9 opening new opportunities for implementing also a larger numbers of knockouts. For example, the genetic manipulation of

***E. coli* has become comparatively simple nowadays with protocols that describe how to implement a knockout within two days.**

5) For yeast in the anaerobic case, coupling can only be achieved for a minority of metabolites. The authors state that this might be due to the mandatory ethanol excretion under these conditions that limit yield. I think in such a case the analysis could also be performed while excluding ethanol production in yield calculations to get a more realistic picture of the possibility of growth coupling in this condition.

This is an interesting suggestion and we have performed a simulation where the ethanol outflow rate is limited to 10 mmol/gDW/h and the product yield is calculated for that fraction of the glucose uptake rate which is not converted into ethanol (assuming that ethanol is produced with optimal yield of 2 mol per mol glucose which may underestimate the amount of glucose that is converted to ethanol). With these settings we could prove that coupling is possible for about 33% of the metabolites at the 10% minimum yield level. However, we have not included these results in the manuscript because with these relaxed yield requirements (which also allows to have very low product yields) they are not directly comparable to the other results and we think it is more consistent to calculate the yield with respect to the entire substrate uptake for all organisms and conditions.

Minor issues:

cMCS are defined on the level of reactions, while genetic implementation requires the knockout of individual genes. It should at least be discussed and exemplified for some cases whether determining cMCSs directly on the gene level would considerably change the conclusions drawn in the paper. For instance, one could test whether growth coupling persists when translating knocked out reactions into the corresponding gene sets and whether those gene sets are still minimal.

Due to the suggestion of the reviewer, we have now also performed the direct calculation of “gene cMCS” for the *E. coli* model (the latter contains a well-established gene-enzyme-reaction association) and have included the results in the manuscript (end of Results section; the Methods section was also extended accordingly). As can be seen, the feasibility of coupling is nearly identical for aerobic conditions while it gets only slightly reduced under anaerobic conditions.

Reviewers' comments:

Reviewer #1 (Remarks to the Author):

The authors have addressed my comments.

Reviewer #2 (Remarks to the Author):

The authors have dutifully met all my requests.

This paper provides substantial evidence that strong growth-coupling is feasible for a majority of metabolites under aerobic circumstances (and for several organisms under anaerobic circumstances as well). Furthermore, it provides the tools to suggest the knockouts needed to provide this strong growth-coupling. I myself am eager to use it in my own research. Therefore, I strongly recommend its publication.

There are a couple of typos that need to be addressed:

Page 10, line 309: "levelsthus" should be "levels, thus"

Page 13, line 418: I think the authors meant "...can be designed through an iterative..." instead of "...can be designed though an interative..."

Reviewer #3 (Remarks to the Author):

Review

In their revision, the authors have responded to the points raised in my prior review. While they have addressed some of my comments and they are addressing a matter of certain importance from a biotechnological perspective I still feel that the results they present are mostly of theoretical rather than practical importance.

While showing that knockout strategies for coupling exist for most metabolites, the manuscript still leaves very much to be desired with respect to the practicability of implementing those knockouts for several reasons:

1) Rather than the question that coupling can be achieved theoretically for arbitrarily high numbers of knockouts it is way more important from a biotechnological perspective to know for how many metabolites coupling can be achieved for a number of knockouts that can be realistically implemented. In my opinion obtaining such numbers should be easily possible by a review of the relevant literature. I would consider such information way more important than a discussion on potential improvements in the ability to implement knockouts with novel technologies. Results should be discussed in the context of a biological feasible number of knockouts rather than assuming that an arbitrarily high number of knockouts can be implemented.

2) It still remains questionable whether the underlying assumptions concerning the excretion of by-products made by the authors are valid. While their argument that *E. coli* will certainly not excrete all possible compounds has some merit, it has been shown previously that *E. coli* does start excreting compounds upon knockouts in central metabolism (in the work I referenced in my last review, Pande et al., ISME J. 2014, 8(5):953-62). In the simplest case blocking a reaction consuming a metabolite will lead to an accumulation of that metabolite. If a transporter of this metabolite exists (or of another upstream metabolite whose concentration probably increases, too), this will lead to an increased excretion of this metabolite. Hence, a production organism might not be able to excrete all compounds for which exchange reaction exists but excretion of some non-canonical products is likely and has already been shown in some cases. In particular, while the authors discuss for how many metabolites coupling can be achieved in the case where excretion reactions are not blocked, they do not provide any additional information with regard to the size of the resulting cut sets (this also relates to my comments concerning the feasibility of in vivo implementation of those knockouts). I think one approach which does not allow for the excretion of all compounds but uses realistic assumptions about which metabolites are potentially excreted would be to consider actual proteomic data, e.g. from the recent Nature Biotechnology publication from the group of Matthias Heinemann. One could define a cut-off on minimal protein counts to define which metabolites are potentially excreted based on that data and repeat calculations for the case in which excretion of metabolites, for which transporters exist in sufficient amounts, is allowed. While this approach certainly requires additional assumptions with respect to the relationship between protein number and fluxes it at least would considerably improve the biological plausibility of the knockout strategies that are derived, at least in the case of *E. coli*. Moreover, this allows to exemplify whether the size of the cut sets changes drastically or only marginally by considering potential excretion reactions.

3) While it might not be possible to prove feasibility of all cut sets due to the computational complexity of the underlying optimization problem there is a trade-off of this argument with the practical feasibility of implementing these knockouts. The authors already exemplify that adding just 10 minutes per run in the case of *E. coli* reduces the mean size of cut sets from 21 reactions to 13 reactions. Though still a large number, knocking out 13 reactions in *E. coli* is much closer to e.g. previous reports from the Sreenc lab reporting on the knockout of 7 reactions for improving the production of ethanol from hexoses and pentoses in *E. coli*. Also their argument concerning scalability of the distributed and multi-threaded computation for solving MILPs by CPLEX is not really of concern as they have to perform calculations for all metabolites. Thus, rather than solving the optimization problem for a single metabolite on several cores or even machines (which certainly impacts scalability), the most efficient approach would be to perform calculations for several metabolites in parallel which should scale almost linearly (i.e. for n CPUs, computation time t would be reduced to t/n). In the present version of the manuscript they have performed computations on just 12 CPUs, which is a very low number considering the computational resources typically available to most bioinformatics groups nowadays (basically the equivalent of three standard desktop PCs).

Response to reviewer comments (2nd round of reviews)

Reviewer #1:

The authors have addressed my comments.

Thanks.

Reviewer #2:

The authors have dutifully met all my requests.

This paper provides substantial evidence that strong growth-coupling is feasible for a majority of metabolites under aerobic circumstances (and for several organisms under anaerobic circumstances as well). Furthermore, it provides the tools to suggest the knockouts needed to provide this strong growth-coupling. I myself am eager to use it in my own research. Therefore, I strongly recommend its publication.

Thanks.

There are a couple of typos that need to be addressed:

Page 10, line 309: "levelsthus" should be "levels, thus"

Page 13, line 418: I think the authors meant "...can be designed through an iterative..." instead of "...can be designed though an interative..."

Thanks, these typos have been fixed.

Reviewer #3:

In their revision, the authors have responded to the points raised in my prior review. While they have addressed some of my comments and they are addressing a matter of certain importance from a biotechnological perspective I still feel that the results they present are mostly of theoretical rather than practical importance. While showing that knockout strategies for coupling exist for most metabolites, the manuscript still leaves very much to be desired with respect to the practicability of implementing those knockouts for several reasons:

1) Rather than the question that coupling can be achieved theoretically for arbitrarily high numbers of knockouts it is way more important from a biotechnological perspective to know for how many metabolites coupling can be achieved for a number of knockouts that can be realistically implemented. In my opinion obtaining such numbers should be easily possible by a review of the relevant literature. I would consider such information way more important than a discussion on potential improvements in the ability to implement knockouts with novel technologies. Results should be discussed in the context of a biological feasible number of knockouts rather than assuming that an arbitrarily high number of knockouts can be implemented.

We still believe that the main merit of our work is the proof that coupling is in principle, i.e. stoichiometrically, feasible for almost all metabolites – a very surprising result, which is, even if further constraints need to be considered for practical implementation, of fundamental importance for rational metabolic engineering (as also emphasized by the other reviewers).

Clearly, a practical implementation of a cut set might involve more specific constraints and requirements, depending on the relevant product and organism. We agree that the cut set size is one important aspect in this regard. We now further extended the discussion on cut sets size, its impact on the feasibility of a calculated intervention strategy and possible ways to reduce the number of knockouts. We now also include a cut set size histogram (for the 50% yield threshold in *E. coli*) in the Supplementary Table which should enable the reader to assess the full spectrum of cut set sizes. Furthermore, we now discuss recent developments in genetic engineering in light of the size distribution of the computed cut sets. In particular, we cite a recent work of Jensen et al. (Scientific Reports, 2015, 5:17874) who presented a technique by which seven gene knockouts can be implemented in *E. coli* within just seven days. It thus seems realistic that 16 reaction knockouts can be introduced in *E. coli* with reasonable effort. With that, we can conclude that more than 75% of all cut sets found for the 50% yield threshold (with extended minimization) would be feasible - which is still a very high percentage. And 16 knockouts will certainly not be the final limit in the future ... but making further statements on the feasible number of knockouts is rather speculative.*

The full paragraph on this discussion now reads as follows:

*The number of knockouts to be implemented is a relevant criterion for assessing the feasibility of a knockout strategy. For a smaller fraction of metabolites, even after spending more time for minimization, we identified very large cut sets. As an example, Supplementary Table 1 shows the histogram of the cut set sizes found for the extended computation for the 50% yield threshold in *E. coli* (where the average cut set size is 12.9; cf. column 50* in Fig. 2). 4.4% of the found intervention strategies would involve more than 20 reaction knockouts which might appear unrealistic. In those cases, as mentioned above, for a single (particular) product of interest, one may drastically increase the computation time to further reduce the cut set size, if possible all the way to the optimum. If the found cut sets are still (too) large, some of the targeted pathways, especially in the anabolism, can often be assumed to have a very low capacity and could therefore be excluded when implementing the knockout strategies, at least in a first attempt. Furthermore, given the ongoing evolution of genome editing techniques, the experimental implementation also of cut sets with a larger number of knockouts can be expected to be feasible, especially in a model organism like *E. coli*. For instance a well-known technique for deleting arbitrary genes in *E. coli* has already been published in the year 2000 which requires about six days to establish a knockout. Recently, a similar technique has been proposed which enabled the implementation of seven gene knockouts in only seven days [Jensen et al., 2015]. Mutant strains with up to 16 reaction knockouts appear therefore not unrealistic anymore, with which more than 75% of the cut sets found in the extended *E.**

coli 50% yield scenario would already become feasible. Finally, the CRISPR-Cas9 system has recently been shown to be a very efficient tool for multiple genetic manipulations, also in more complex organisms, and its particular potential for metabolic engineering has been emphasized.

2) It still remains questionable whether the underlying assumptions concerning the excretion of by-products made by the authors are valid. While their argument that *E. coli* will certainly not excrete all possible compounds has some merit, it has been shown previously that *E. coli* does start excreting compounds upon knockouts in central metabolism (in the work I referenced in my last review, Pande et al., ISME J. 2014, 8(5):953-62).

*The paper cited by the reviewer reports about the excretion of amino acids by *E. coli* after deleting certain genes. In fact, the gene deletions (which were suggested by a modeling technique similar to the cut sets used in our paper) were introduced with the goal to overproduce these amino acids. Hence, this study is rather a positive than a counter example because model predictions led to the desired and intended (predicted, not unexpected) excretion of amino acids.*

In the simplest case blocking a reaction consuming a metabolite will lead to an accumulation of that metabolite. If a transporter of this metabolite exists (or of another upstream metabolite whose concentration probably increases, too), this will lead to an increased excretion of this metabolite. Hence, a production organism might not be able to excrete all compounds for which exchange reaction exists but excretion of some non-canonical products is likely and has already been shown in some cases. In particular, while the authors discuss for how many metabolites coupling can be achieved in the case where excretion reactions are not blocked, they do not provide any additional information with regard to the size of the resulting cut sets (this also relates to my comments concerning the feasibility of in vivo implementation of those knockouts).

*As suggested by the reviewer, we now included a note in the manuscript that the average size of the cut sets found for *E. coli* in the case where all excretion reactions are open increases by seven reaction knockouts.*

I think one approach which does not allow for the excretion of all compounds but uses realistic assumptions about which metabolites are potentially excreted would be to consider actual proteomic data, e.g. from the recent Nature Biotechnology publication from the group of Matthias Heinemann. One could define a cut-off on minimal protein counts to define which metabolites are potentially excreted based on that data and repeat calculations for the case in which excretion of metabolites, for which transporters exist in sufficient amounts, is allowed. While this approach certainly requires additional assumptions with respect to the relationship between protein number and fluxes it at least would considerably improve the biological plausibility of the knockout strategies that are derived, at least in the case of *E. coli*. Moreover, this allows to exemplify whether the size of the cut sets changes drastically or only marginally by considering potential excretion reactions.

*We do not believe that the approach suggested by the reviewer will increase the plausibility of the calculated cut sets as it would be highly speculative. The data in the cited paper by Heinemann et al. contains proteome data for *E. coli* wild type strains only (measured under different growth conditions; e.g., minimal media with an excess of different carbon and energy sources, glucose-*

limited chemostat cultures with varying growth rates, growth on glucose excess with different stress conditions, growth on complex medium). We do not see how these data can bring evidence which metabolites will be excreted by the respective *E. coli* mutant strains where several knockouts have been introduced. For example, assume that a transporter for a particular metabolite (different from the standard exchange metabolites considered in our computations) is measured with an abundance above a chosen threshold. Since *E. coli* wild type does not normally excrete other metabolites than the standard products this demonstrates that mere existence of a transporter does not prove excretion of the respective metabolite. Why should we then assume in the calculations that this transporter is active in the mutant strains if it is not active in the wild type (despite significant amounts of the transporter measured in the wild type proteome)? This is not a reasonable assumption, even more because certain transporters might be expressed (or repressed) just as a consequence of the introduced gene knockouts in the designed strains, which will not be predictable by the proteome of the wild type strain.

As can be seen by the long paragraph in the discussion section in the manuscript, we are certainly aware of the fact that the (non-predictable) excretion of certain metabolites adds uncertainty to the calculated cut sets and feasibility of coupling (which, by the way, is not specific to our approach but an issue of all strain design methods used to find interventions for growth-coupled product synthesis). However we feel that we gave a thorough discussion of this point and that we properly tackled this problem in two directions:

1) We consider two extremes: the best-case (only standard excretion reactions are active) and the worst-case (all exchange reactions active, e.g., 285 (!) in *E. coli*). The reality will be somewhere between these two extremes, though it will certainly be much closer to the best-case as normally only some of the potential exchange metabolites will be excreted, not all 285 simultaneously. Since we have shown that even in the worst-case a significant percentage of all metabolites can be coupled to growth (between 36% and 81%, depending on the organism) the main conclusion of our paper remains valid even in the most unfavorable case. The feasibility for the worst-case will even be better if we also allowed knockouts of transport reactions (genes for transporters): if all exchange reactions can be targeted, then the worst-case would yield the same results on feasibility as the best-case (but potentially require more knockouts).

2) We also provided a practical solution for cases where a production strain constructed from a cut set excretes a non-conventional exchange metabolite: When, after experimental implementation of some knockouts of a calculated cut set, a metabolite is excreted whose outflow was not considered in the model before, it is possible to modify the model accordingly and then to recalculate and adapt the current cut set(s) in order to get intervention strategies which additionally suppress the unwanted excretion. In this manner, a production strain can be designed through an iterative cycle of calculation and experiment as was recently demonstrated for high-yield itaconic acid synthesis in *E. coli* [Harder et al. *Metabolic Engineering*, 2016, 38: 29 - 37].

3) While it might not be possible to prove feasibility of all cut sets due to the computational complexity of the underlying optimization problem there is a trade-off of this argument with the practical feasibility of implementing this knockouts. The authors already exemplify that adding just 10 minutes per run in the case of *E. coli* reduces the mean size of cut sets from 21 reactions to 13 reactions.

Here we would like to mention that we added 10 minutes per metabolite which required additional 8 days computation time on our system (see the Table in Figure 2).

Though still a large number, knocking out 13 reactions in *E. coli* is much closer to e.g. previous reports from the Srienc lab reporting on the knockout of 7 reactions for improving the production of ethanol from hexoses and pentoses in *E. coli*. Also their argument concerning scalability of the distributed and multi-threaded computation for solving MILPs by CPLEX is not really of concern as they have to perform calculations for all metabolites. Thus, rather than solving the optimization problem for a single metabolite on several cores or even machines (which certainly impacts scalability), the most efficient approach would be to perform calculations for several metabolites in parallel which should scale almost linearly (i.e. for n CPUs, computation time t would be reduced to t/n). In the present version of the manuscript they have performed computations on just 12 CPUs, which is a very low number considering the computational resources typically available to most bioinformatics groups nowadays (basically the equivalent of three standard desktop PCs).

*We agree with the reviewer that the optimal strategy to distribute the computation might be to use one core or (cluster node) per metabolite (rather than parallelizing the computation of each metabolite). However, even in this case, due to the combinatorial complexity of the problem, it is not guaranteed at all that the optimal (minimal MCS) solution will be found for a given metabolite. To exemplify this, we chose one metabolite of *E. coli* where only a large cut set (with more than 30 interventions) was found for coupling. We started an optimization on one of our cluster nodes with the goal to find the MCS with minimum size (enforcing growth-coupled production of this metabolite) within a time limit of one week. We stopped the calculation after 7 days as the solver did not succeed in finding the optimal solution. This example shows that for the “difficult” cases (metabolites with larger cut sets) it will be hard or even infeasible to find the absolute minimum solution (because the search tree within the MILP solver becomes huge). For this reason, we decided to relax the problem to allow also MCS with larger sizes to be found which still prove feasibility of coupling.*

*Again, the fact that the cut sets do not all have minimal size does not affect the main result of this paper, namely that growth-coupled product synthesis is in principle feasible for almost all metabolites. Despite the fact that we cannot guarantee minimum size of the cut sets for most metabolites, we also note that the MCSs obtained from the additional optimization performed for the *E. coli* 50% scenario (the 50* column in the Table in Figure 2) are likely to be close to the MCS with minimum size. As already mentioned above, in this scenario, spending 10 minutes extra for each metabolite (requiring 8 days for the additional computation) we obtained an average MCS size of 12.9 (compared to 20.6 in the first run). Spending additional 10 minutes (20 minutes in total for each metabolite; requiring again ~8 additional days on a single cluster node) we obtained MCS with an average cut set size of 12.6 indicating that there was only a minor further reduction in the MCS size with this additional calculation. Nevertheless, there was only a minor increase of the number of metabolites for which minimality could be proven (73 instead of 68). All these results underline that a full decrease of the MCS size down to the optimum (or the proof that the optimum has been found) requires an exponential increase in the computational resources and is therefore only worthwhile to perform when an experimental implementation of the MCS for a particular product is indeed planned.*

References:

Jensen SI, Lennen RM, Herrgård MJ, Nielsen AT (2015) Seven gene deletions in seven days: Fast generation of *Escherichia coli* strains tolerant to acetate and osmotic stress. *Scientific Reports* 5:17874.

REVIEWERS' COMMENTS:

Reviewer #4 (Remarks to the Author):

With the additional revisions the authors have suitably addressed my concerns, especially concerning the optimality of the solutions they identify and the practicability of the implementation of the predicted knockouts.

Response to reviewer comments (3nd round of reviews)

Reviewer #4:

With the additional revisions the authors have suitably addressed my concerns, especially concerning the optimality of the solutions they identify and the practicability of the implementation of the predicted knockouts.

We thank the reviewer for the positive comment and have not introduced further changes in the manuscript.